# The pitfalls of using Gaussian Process Regression for normative modeling

**Bohan Xu**[1,2]*, **Rayus Kuplicki**[1], **Sandip Sen**[2], **Martin P. Paulus**[1,3,4]

**1** Laureate Institute for Brain Research, Tulsa, OK, United States of America, **2** Department of Computer Science, Tandy School of Computer Science, University of Tulsa, Tulsa, OK, United States of America, **3** Department of Community Medicine, Oxley College of Health Sciences, University of Tulsa, Tulsa, OK, United States of America, **4** Department of Psychiatry, School of Medicine, University of California San Diego, San Diego, CA, United States of America

* bxu@laureateinstitute.org

**Data Availability Statement:** The data is generated from a python script, and the python script is available on the Github (https://github.com/nidaye1999/normative-model-GPR).

**Funding:** This research was supported by the Laureate Institute for Brain Research and the

## Abstract

Normative modeling, a group of methods used to quantify an individual's deviation from some expected trajectory relative to observed variability around that trajectory, has been used to characterize subject heterogeneity. Gaussian Processes Regression includes an estimate of variable uncertainty across the input domain, which at face value makes it an attractive method to normalize the cohort heterogeneity where the deviation between predicted value and true observation is divided by the derived uncertainty directly from Gaussian Processes Regression. However, we show that the uncertainty directly from Gaussian Processes Regression is irrelevant to the cohort heterogeneity in general.

## Introduction

In case-control studies, participants are assigned labels and classified into one or more categories based on their similarities or common criteria, with little consideration for the heterogeneity within each cohort. Meanwhile, normative modeling is becoming increasingly popular. In a normative model, each observation is quantified as a normalized deviation with respect to the cohort heterogeneity. The growth chart [1, 2] is an example normative model as shown in Fig 1, where a series of percentile curves (normalized deviation) illustrate the distribution of selected body measurements in children. Another widely-used measure for normalized deviation is the $z$-score, which is calculated by dividing the difference between an observation and the reference model, i.e., residual, by a standard deviation that represents local heterogeneity and assumes residuals are Gaussian distributed locally.

The uncertainty sometimes can be classified into two categories: epistemic and aleatoric uncertainties. Epistemic uncertainty is known as systematic uncertainty and is due to things one could in principle know but do not in practice; aleatoric uncertainty is known as statistical uncertainty and is representative of unknowns that differ each time we run the same experiment [4]. Epistemic uncertainty is often introduced by the limited dataset size and can be reduced by adding more observations. On the other hand, aleatoric uncertainty represents a character of heterogeneity in the underlying distribution itself which is unrelated to

National Institute of General Medical Sciences (P20GM121312, MP, RK).

**Fig 1. Weight-for-age boys: Birth to 2 years.** Reprinted from [3] under a CC BY license, with permission from World Health Organization, original copyright (2021). The percentiles show the distribution of weights in boys form birth to 2 years. Black dots: observations; red error bars: epistemic uncertainty; blue curly brackets: aleatoric uncertainty.

sample size, so it cannot be reduced by modifying the dataset, and this is the heterogeneity a normative model should measure. As shown in Fig 1, larger number and density of data points (black dots) reduce the epistemic uncertainty (red error bars), while the aleatoric uncertainty (blue curly brackets) is unrelated to the sample size or distribution. The confidence intervals obtained from most statistical tests and advanced machine learning models only capture epistemic uncertainty, while a normative model is designed to capture the aleatoric uncertainty.

Gaussian Process Regression (GPR) has been widely used in many domains. Schulz et al. [5] presented a tutorial on the GPR with the mathematics behind the model as well as several applications to real-life datasets/problems. Tonner et al. [6] developed a GPR based model and testing framework to capture the microbial population growth and shown their proposed approach outperformed primary growth models. Banerjee et al. [7] and Raissi et al. [8] introduced two novel approaches to improve the efficiency of GPR in "big data" problems.

However, some previous research implemented the GPR as a normative modeling approach and utilized the derived prediction variance to model the cohort heterogeneity. Ziegler et al. [9] attempted to build a normative model for diagnosing mild cognitive impairment and Alzheimer's disease based on the normalized deviation of predicted brain volume from GPR. Marquand et al. [10] used delay discounting as covariates and reward-related brain activity derived from task Functional Magnetic Resonance Imaging (fMRI) as the target variable with GPR and extreme value statistics to identify the participants with Attention-Deficit/Hyperactivity Disorder (ADHD). Wolfers et al. [11] investigated the deviation of brain volume in an ADHD cohort from healthy control group (HC) with respect to age and gender, and they also explored the heterogeneous phenotype of brain volume for schizophrenia and bipolar disorder with GPR [12]. Zabihi et al. [13] studied Autism Spectrum Disorder (ASD) regarding the deviation of cortical thickness via a similar methodology.

In this paper, we introduce some background knowledge related to GPR. We then present a rigorous mathematical derivation and several examples to demonstrate that the variance from GPR cannot be used in a normative model alone. In the last section, we discuss the difficulties and disadvantages of modeling the cohort heterogeneity by modifying original GPR variance, and a misunderstanding existed in previous research.

## Materials and methods

### Gaussian Process Regression

The relation between the observation and the predictive model usually can be expressed as

$$y = f(\boldsymbol{x}) + \varepsilon, \tag{1}$$

where $y$ is the observation (output), $f(\cdot)$ represents the predictive model, $\boldsymbol{x}$ is a vector of independent variables (input) corresponding to the output $y$, and $\varepsilon$ is the noise term which follows a normal distribution $\varepsilon \sim \mathcal{N}\left(0, \sigma_{\text{noise}}^2\right)$. Gaussian Process Regression (GPR) assumes a zero-mean normal distribution over the predictive model

$$f(\cdot) \sim \mathcal{N}(0, k(\cdot, \cdot)), \tag{2}$$

where $k(\cdot, \cdot)$ is some covariance (kernel) function. Given the training set input $\boldsymbol{X}$ and testing set input $\boldsymbol{X}_*$, since both of them follow the same distribution, we have

$$f\left(\begin{bmatrix} \boldsymbol{X} \\ \boldsymbol{X}_* \end{bmatrix}\right) \sim \mathcal{N}\left(\boldsymbol{0}, \begin{bmatrix} \boldsymbol{K}(\boldsymbol{X}, \boldsymbol{X}) & \boldsymbol{K}(\boldsymbol{X}, \boldsymbol{X}_*) \\ \boldsymbol{K}(\boldsymbol{X}_*, \boldsymbol{X}) & \boldsymbol{K}(\boldsymbol{X}_*, \boldsymbol{X}_*) \end{bmatrix}\right). \tag{3}$$

According to the Eq 1, the observation follows the summation of these two normal distributions

$$\begin{bmatrix} \boldsymbol{y} \\ \boldsymbol{y}_* \end{bmatrix} = f\left(\begin{bmatrix} \boldsymbol{X} \\ \boldsymbol{X}_* \end{bmatrix}\right) + \begin{bmatrix} \varepsilon \\ \varepsilon_* \end{bmatrix} \sim \mathcal{N}\left(\boldsymbol{0}, \begin{bmatrix} \boldsymbol{K}(\boldsymbol{X}, \boldsymbol{X}) + \Sigma_{\text{train}}^2 & \boldsymbol{K}(\boldsymbol{X}, \boldsymbol{X}_*) \\ \boldsymbol{K}(\boldsymbol{X}_*, \boldsymbol{X}) & \boldsymbol{K}(\boldsymbol{X}_*, \boldsymbol{X}_*) + \Sigma_{\text{test}}^2 \end{bmatrix}\right), \tag{4}$$

where $\Sigma_{\text{train}}^2$ and $\Sigma_{\text{test}}^2$ are two square diagonal matrices that represent the variance of observation noise in training and testing sets, and all diagonal elements of $\Sigma_{\text{train}}^2$ and $\Sigma_{\text{test}}^2$ are identical and equal to $\sigma_{\text{noise}}^2$. By the rules of conditional Gaussian distribution, the prediction of testing set $\boldsymbol{y}_*$ follows a normal distribution $\boldsymbol{y}_* \sim \mathcal{N}\left(\mu_*, \Sigma_*^2\right)$, where $\mu_*$ and $\Sigma_*^2$ are defined as [14, 15]

$$\mu_* = \boldsymbol{K}(\boldsymbol{X}_*, \boldsymbol{X})[\boldsymbol{K}(\boldsymbol{X}, \boldsymbol{X}) + \Sigma_{\text{train}}^2]^{-1}\boldsymbol{y}, \tag{5a}$$

$$\Sigma_*^2 = \boldsymbol{K}(\boldsymbol{X}_*, \boldsymbol{X}_*) + \Sigma_{\text{test}}^2 - \boldsymbol{K}(\boldsymbol{X}_*, \boldsymbol{X})[\boldsymbol{K}(\boldsymbol{X}, \boldsymbol{X}) + \Sigma_{\text{train}}^2]^{-1}\boldsymbol{K}(\boldsymbol{X}, \boldsymbol{X}_*). \tag{5b}$$

### Kernel trick

Similar to Support Vector Machines (SVM), the kernel trick can also be implemented with GPR to project the input of data from the original space into a same or higher dimensional feature space via some mapping function $z(\cdot)$. Given a pair of inputs $(\boldsymbol{x}_1, \boldsymbol{x}_2)$, the kernel function calculates the inner product of the coordinates in the feature space, i.e., $k(\boldsymbol{x}_1, \boldsymbol{x}_2) = z(\boldsymbol{x}_1)z(\boldsymbol{x}_2)^T$ [16, 17]. The kernel trick avoids the expensive computation of calculating the coordinate in the feature space for each input. We use the linear kernel and Radial Basis Function kernel (RBF) as examples to illustrate this advantage.

**Linear kernel.** The linear kernel is non-stationary and the simplest kernel, which is defined as

$$k(\boldsymbol{x}_1, \boldsymbol{x}_2) = \boldsymbol{x}_1\boldsymbol{x}_2^T, \tag{6a}$$

$$z(\boldsymbol{x}) = \boldsymbol{x}, \tag{6b}$$

where the input is projected into a feature space according to Eq 6b, and the feature space is the original space.

**Radial Basis Function kernel.** The RBF kernel is a stationary kernel, which is also widely used and defined as [17]

$$k(\boldsymbol{x}_1, \boldsymbol{x}_2) = e^{-\frac{\|\boldsymbol{x}_1 - \boldsymbol{x}_2\|^2}{2l^2}}, \tag{7a}$$

$$z(\boldsymbol{x}) = \left[\frac{e^{-\frac{\|\boldsymbol{x}\|^2}{2l^2 j}}}{\sqrt{l^{2j} j!}} \sqrt{\frac{j!}{n_1! \cdots n_k!}} x_1^{n_1} \cdots x_k^{n_k}\right]_{j=0,\cdots,\infty, \sum_{i=1}^{k} n_i = j}, \tag{7b}$$

where $l$ is a free scaling parameter. The RBF kernel projects the input from the original space onto an infinite dimensional feature space where the mapping is defined by Eq 7b. It is impossible to exactly compute the coordinates in an infinite dimensional space, while Eq 7a still allows straightforward computation of the inner product for coordinate pairs in that feature space.

**Matérn and Rational-Quadratic kernels.** Matérn kernel is a generalization of the RBF kernel which is defined as [14]

$$k(\boldsymbol{x}_1, \boldsymbol{x}_2) = \frac{2^{1-v}}{\Gamma(v)}\left(\frac{\sqrt{2v}}{l}\|\boldsymbol{x}_1 - \boldsymbol{x}_2\|\right)^v K_v\left(\frac{\sqrt{2v}}{l}\|\boldsymbol{x}_1 - \boldsymbol{x}_2\|\right), \tag{8}$$

where the parameter $v$ controls the smoothness of the function, $\Gamma(\cdot)$ refers to the gamma function, and $K_v(\cdot)$ represents modified Bessel function. Rational-Quadratic kernel is another kernel based on the RBF kernel, which is given by

$$k(\boldsymbol{x}_1, \boldsymbol{x}_2) = \left(1 + \frac{\|\boldsymbol{x}_1 - \boldsymbol{x}_2\|^2}{2\alpha l^2}\right)^{-\alpha}, \tag{9}$$

where $\alpha$ is a scale mixture parameter. The Rational-Quadratic kernel can be considered as an infinite sum of RBF kernels with different length-scales $l$ [18].

## Estimated uncertainty for GPR

One benefit of using GPR to build a data-driven model is the predictions are associated with the derived variances as shown in Eq 5. However, we need to emphasize that this variance is only related to the kernel function $k(\cdot, \cdot)$ and distribution/coordinate of training set input $\boldsymbol{X}$, i.e., it cannot be utilized in a normative model approach alone to capture the variance introduced by the conditional distribution $Var(y|\boldsymbol{x})$.

We better illustrate and verify this statement through simplifying the Eq 5b. Since any kernel function $k(\cdot, \cdot)$ can be written as the inner product of a coordinate pair in the feature space by some mapping function $z(\cdot)$, we present our derivation in a general format. We define a variable $\boldsymbol{x}_*$ which represents a testing input, then Eq 5b can be written as

$$\Sigma_*^2(\boldsymbol{x}_*)$$
$$= k(\boldsymbol{x}_*, \boldsymbol{x}_*) + \sigma_{\text{test}}^2 - \boldsymbol{k}(\boldsymbol{x}_*, \boldsymbol{X})[\boldsymbol{K}(\boldsymbol{X}, \boldsymbol{X}) + \Sigma_{\text{train}}^2]^{-1}\boldsymbol{k}(\boldsymbol{X}, \boldsymbol{x}_*) \tag{10a}$$
$$= \boldsymbol{z}(\boldsymbol{x}_*)\boldsymbol{z}(\boldsymbol{x}_*)^T + \sigma_{\text{test}}^2 - \boldsymbol{z}(\boldsymbol{x}_*)\boldsymbol{Z}(\boldsymbol{X})^T[\boldsymbol{Z}(\boldsymbol{X})\boldsymbol{Z}(\boldsymbol{X})^T + \Sigma_{\text{train}}^2]^{-1}\boldsymbol{Z}(\boldsymbol{X})\boldsymbol{z}(\boldsymbol{x}_*)^T. \tag{10a}$$

Applying Singular Value Decomposition (SVD) on $\sum_{\text{train}}^{-1} Z(X) = U \sum V^T$, Eq 10 is reformulated as (the detailed derivation is presented in S1 Appendix)

$$
\begin{aligned}
\Sigma_*^2(x_*) & \\
= \; & z(x_*)z(x_*)^T + \sigma_{\text{test}}^2 - z(x_*)Z(X)^T[Z(X)Z(X)^T + \Sigma_{\text{train}}^2]^{-1}Z(X)z(x_*)^T \\
= \; & \underbrace{z(x_*)V[I - \Sigma^T(\Sigma\Sigma^T + I)^{-1}\Sigma]V^Tz(x_*)^T}_{\text{quadratic term}} + \underbrace{\sigma_{\text{test}}^2}_{\text{constant}}.
\end{aligned}
\tag{11}
$$

After simplification, the variance is reformulated as Eq 11, which is a summation of a quadratic term for $z(x_*)$ and a constant represents the noise, $\Sigma$ and $V$ are constant matrices where the values are fully depended on training input $X$, training noise $\Sigma_{\text{train}}$, and mapping function $z(\cdot)$ or kernel function $k(\cdot, \cdot)$.

## Modification of uncertainty from GPR

Regarding Eq 11, the variance calculated via Eq 5b is purely depended on kernel function and training data input, thus it is only able to capture the epistemic uncertainty which could be reduced by modifying or adding training data. The derived variance from GPR could be extended to model the heterogeneity $Var(y|x)$ for a normative model by adding an aleatoric variance term into Eq 5b

$$
\begin{aligned}
Var(y_*|X_*) = \; & \underbrace{K(X_*, X_*) + \Sigma_{\text{test}}^2 - K(X_*, X)[K(X, X) + \Sigma_{\text{train}}^2]^{-1}K(X, X_*)}_{\text{epistemic uncertainty}} \\
& + \underbrace{\Sigma_{\text{aleatoric}}^2(X_*)}_{\text{aleatoric uncertainty}},
\end{aligned}
\tag{12}
$$

where $\sum_{\text{aleatoric}}^2(X_*)$ represents the data character of heterogeneity in output at given locations on the input space. This formula, however, is not implemented in any previous research as we know and we will discuss the difficulties and disadvantages in estimating the aleatoric uncertainty later.

## Results

We apply the unmodified GPR (Eq 5) on several synthetic datasets where both input $x$ and output $y$ are one dimensional to facilitate visualization. Although the presented results are based on one dimensional input $x$, they are generalizable to any dimensional input. We illustrate the characteristics of the four kernels mentioned above, but we mainly focus on the linear and RBF kernels. We also present the results of two scenarios with known and unknown noise levels.

## Dataset

Four synthetic datasets are generated and plotted in Fig 2, and each of them contains 1000 points with a noise level of $\sigma_{\text{noise}} = 0.05$. Four other undersampled datasets are plotted in Fig 3, each of which contains $1000 \times 5\% = 50$ points.

In Dataset 1, both input $X$ and output $y$ follow a Gaussian distribution $\mathcal{N}(0, 1^2)$ and are correlated with a Pearson coefficient of 0.75. Dataset 2 is transformed from Dataset 1, which moves the set of points where $x \geq 0$ in Dataset 1 along the line $y = x$ until the maximum input in that set equals 0, and moves the remaining points where $x < 0$ in Dataset 1 until the

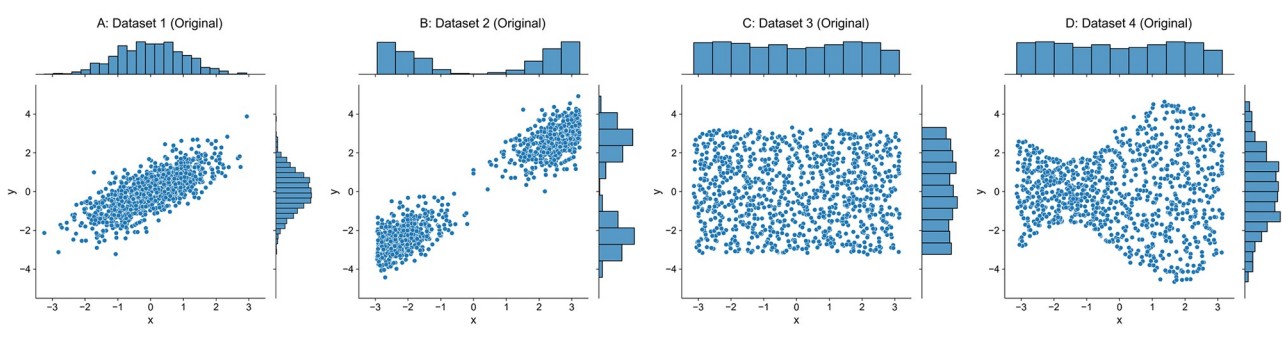

**Fig 2. Original datasets.**

minimum input is 0. Dataset 3 has input $X$ and output $y$ uniformly distributed over a half-open interval $[-\pi, \pi]$. Output $y$ of Dataset 4 is obtained by multiplying a factor function over output $y$ from Dataset 3, which is defined as $f(x) = \sin(x)/2 + 1$ and $x$ is the corresponding input. We should note that the inputs $X$ of original Datasets 3–4 are exactly same as shown in Fig 2C and 2D, and the inputs $X$ of corresponding undersampled Datasets 3–4 are also identical as shown in Fig 3C and 3D.

## GPR with known noise level

**Linear kernel.** The regression surface of GPR with linear kernel is a hyperplane and the variance is a quadric hypersurface defined by Eq 11 in feature/original spaces, where the hyperplane always passes the origin, the variance is a function only with respect to the coordinate of testing input $x_*$ and a unique minimum is located at $x^* = \mathbf{0}$. Figs 4 and 5 present results for GPR with linear kernel on the one dimensional synthetic datasets, where top sub-figures plot the reference models/predictions (red lines) overlapped on the data (blue dots), middle sub-figures show the derived variances (blue curves) across the original input space, and the bottom sub-figures shows the corresponding "$z$-score" for training set which is computed via Eq 13 if the residual ($y - y_{\text{reference}}$) is mistakenly normalized by standard deviation $\Sigma$ directly from GPR (Eq 5b).

$$z-\text{score} = \frac{y - y_{\text{reference}}}{\Sigma} \tag{13}$$

The mapping function of linear kernel projects an input to itself (Eq 6b). For one dimensional input, $\Sigma$ in Eq 11 is an $m \times 1$ matrix, where $m$ is the size of training set. The only non-

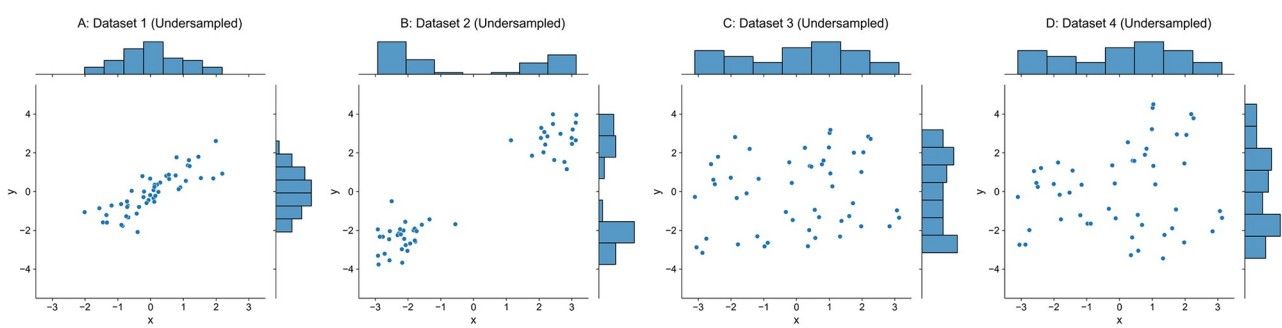

**Fig 3. Undersampled datasets.**

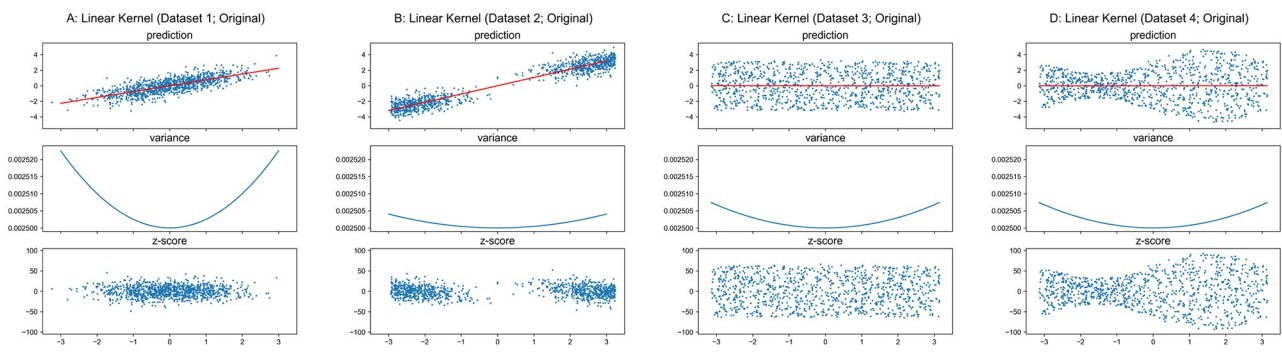

**Fig 4. GPR with linear kernel on original datasets.**

zero element $\Sigma_{1,1}$ equals the only non-zero singular value $\sigma$ of $\sum_{\text{train}}^{-1}X$, and $V$ is a $1 \times 1$ identity matrix. Therefore, Eq 11 can be further reduced to

$$
\begin{aligned}
\Sigma_*^2(x_*) &= [1 - \Sigma^T(\Sigma\Sigma^T + I)^{-1}\Sigma]x_*^2 + \sigma_{\text{test}}^2 \\
&= \left(1 - \frac{\sigma^2}{\sigma^2 + 1}\right)x_*^2 + \sigma_{\text{test}}^2 \\
&= \frac{x_*^2}{\sigma^2 + 1} + \sigma_{\text{test}}^2.
\end{aligned}
\tag{14}
$$

As shown in Figs 4 and 5, the variance is a univariate function of coordinate of the testing input $x^*$ where the shape is a quadratic curve, and the global minimum is always located at $x^* = 0$ with a value of $\sigma_{\text{test}}^2 = 0.05^2$ as Eq 14 formulated. The result of GPR with linear kernel presents a good example which illustrates the derived variance from GPR does not model the conditional variance $Var(y|x)$, thus corresponding $z$-score cannot be utilized as a normalized deviation in a normative model.

As previously mentioned, the predicted variance for testing set from GPR only depends on the training set input and the kernel function. As the original as well as the undersampled Datasets 3–4 have identical inputs $X$, the variance curves in Figs 4C, 4D, 5C and 5D are respectively identical.

**RBF kernel.** Unlike the linear kernel, RBF kernel mapping function (Eq 7b) defines a feature space which is different from the original space. Regarding the original space, the

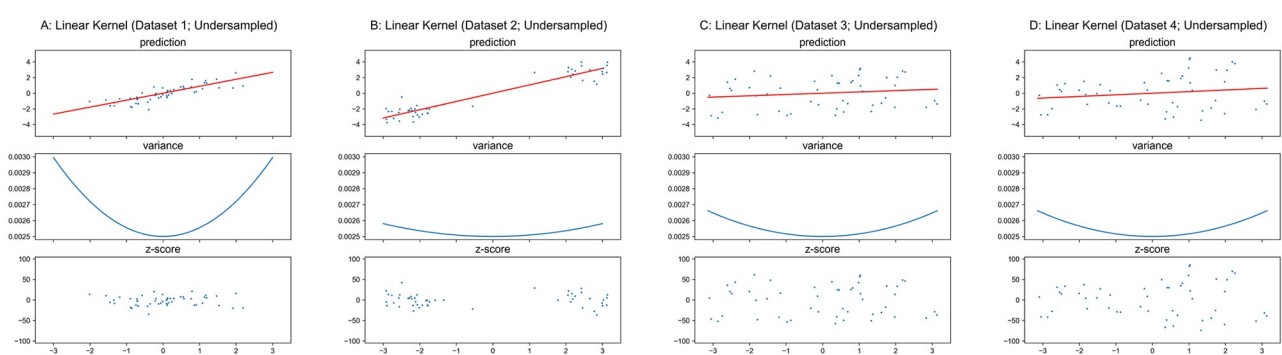

**Fig 5. GPR with linear kernel on undersampled datasets.**

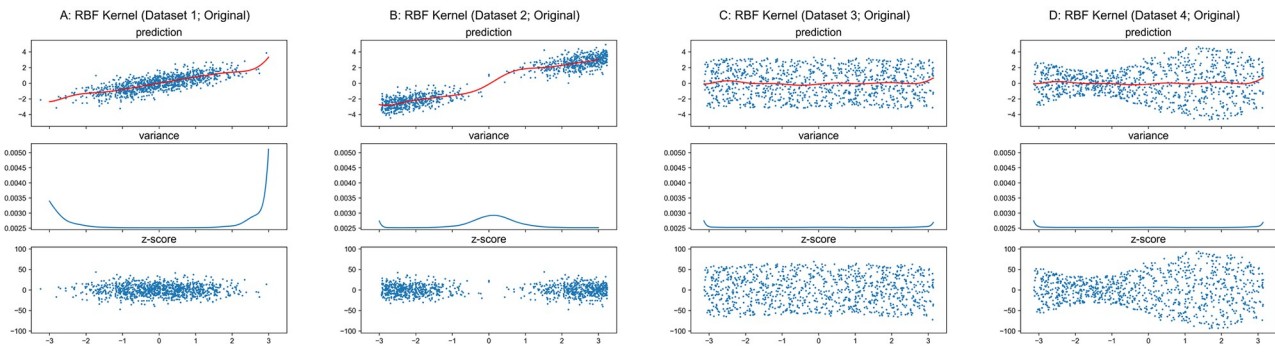

**Fig 6. GPR with RBF kernel on original datasets.**

regression surface is no longer a hyperplane and the variance is no more a quadric hypersurface for the RBF kernel, although regression surface is always a hyperplane and variance is always a quadric hypersurface for any kernels in the feature space. Because the mapping function of RBF kernel is very complicated, we only briefly describe the characteristics of the regression surface and variance in the original space. For a test input $x^*$, the prediction is a summation of discounted outputs of all training points where each corresponding discount factor is determined by the Euclidean distance between $x^*$ and that training input, and the predicted value converges to 0 if $x^*$ is far away from all training inputs. On the other hand, the variance depends only on the density of training inputs at $x^*$, and higher density results in lower variance. Therefore, the variance of GPR with RBF kernel is related of the relative location to the training inputs rather than the absolute location specified by coordinate.

The results for GPR with RBF kernel applied to these synthetic datasets are shown in Figs 6 and 7. We should note that the value of hyper-parameter $l$ in the RBF kernel function (Eq 7a) does not affect the main idea of this paper, thus we used a fixed value of 1.0 instead of utilizing hyper-parameter optimization in this section. As shown in Figs 6 and 7, the variance is unrelated to the conditional variance $Var(y|x)$. Therefore, $z$-scores based on this model do not represent normalized deviation. However, unlike the quadratic curves whose unique minimum is always located at $x = 0$ in Figs 4 and 5 for linear kernel, the variance function of GPR with RBF kernel regarding the original input space is related to the distribution of training input $X$. The denser inputs at the middle of Dataset 1 and two ends of Dataset 2 lead to lower variances at those locations in Fig 6A and 6B, while the uniformly distributed inputs of Datasets 3–4 result in relatively flat curves in and Fig 6C and 6D. According to Eq 7a and given an arbitrary input $x^*$, the RBF kernel function returns a larger value for a point in $X$ that is closer to $x^*$, and $k(x^*, X)$ and $k(X,x^*)$ have more large elements if $x^*$ is close to more points in $X$. Due to $K(X, X) + \sum_{\text{train}}^2$ is a symmetric positive definite matrix, both result in the decrease of the value for Eq 10a, i.e., to smaller variance.

Similar to the result for the linear kernel, the theoretical minimum of variance is $\sigma_{\text{test}}^2 = 0.05^2$, and the variance curves are exactly identical in Figs 6C, 6D, 7C and 7D respectively.

**Matérn and Rational-Quadratic kernels.** The properties of Matérn and Rational-Quadratic kernels are similar to the RBF kernel. Therefore, we only present the results in Figs 8–11 for these two kernels without the detailed analysis. Similar to RBF kernel, the hyper-parameters of Matérn and Rational-Quadratic kernels are also fixed in this section, where $v = 1.5$ for Matérn kernel, $\alpha = 1.0$ for Rational-Quadratic kernel, and $l = 1.0$ for both kernels. The

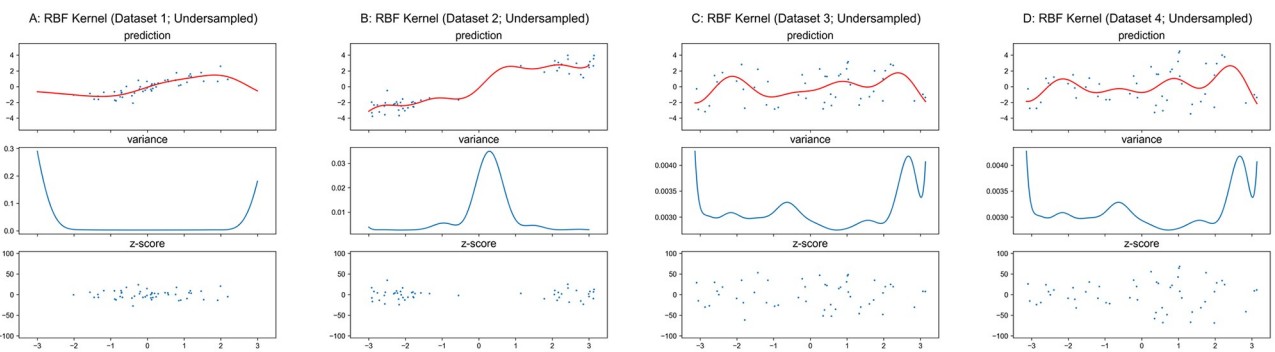

**Fig 7. GPR with RBF kernel on undersampled datasets.** Scales of Y-axis for variance plots are different.

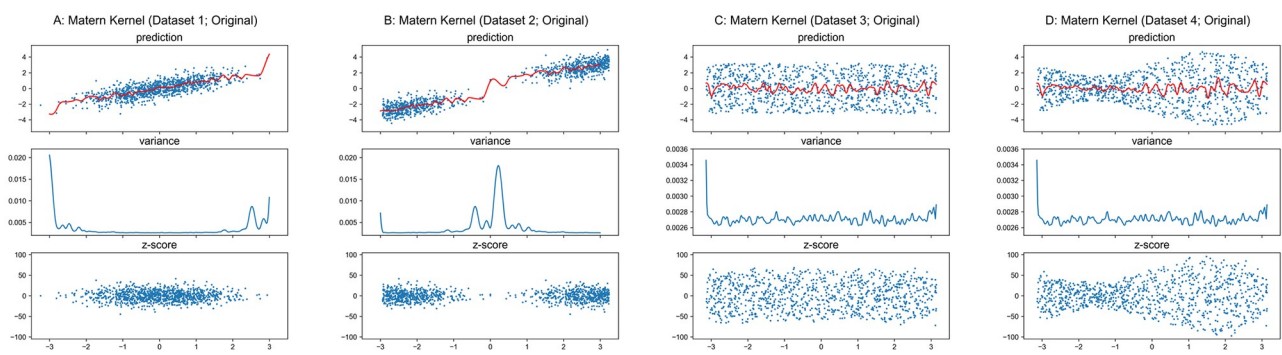

**Fig 8. GPR with matérn kernel on original datasets.** Scales of Y-axis for variance plots are different.

variance curves shown in Figs 8C, 8D, 9C, 9D, 10C, 10D, 11C and 11D are exactly identical, respectively.

## GPR with unknown noise level

The noise level can be included as a hyper-parameter when it is unknown. However, the derived variance from GPR still does not model the heterogeneity $Var(y|\boldsymbol{x})$, although it could be a good approximation in some special cases.

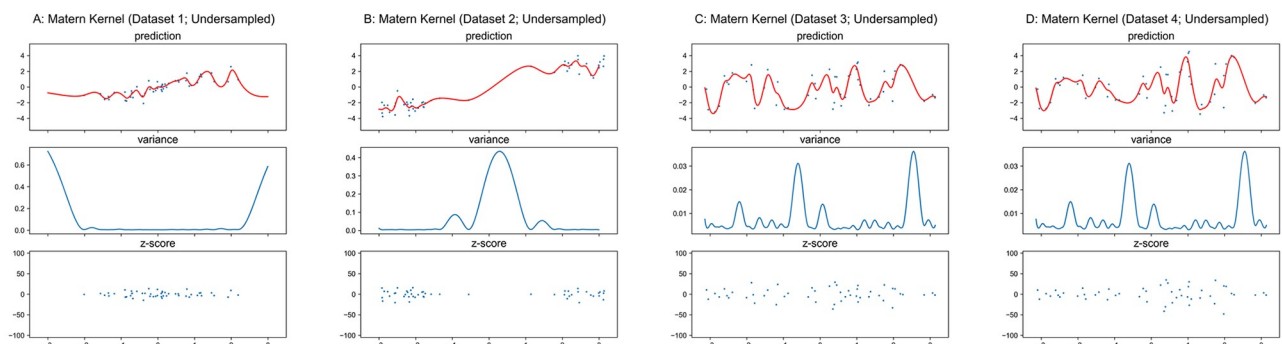

**Fig 9. GPR with matérn kernel on undersampled datasets.** Scales of Y-axis for variance plots are different.

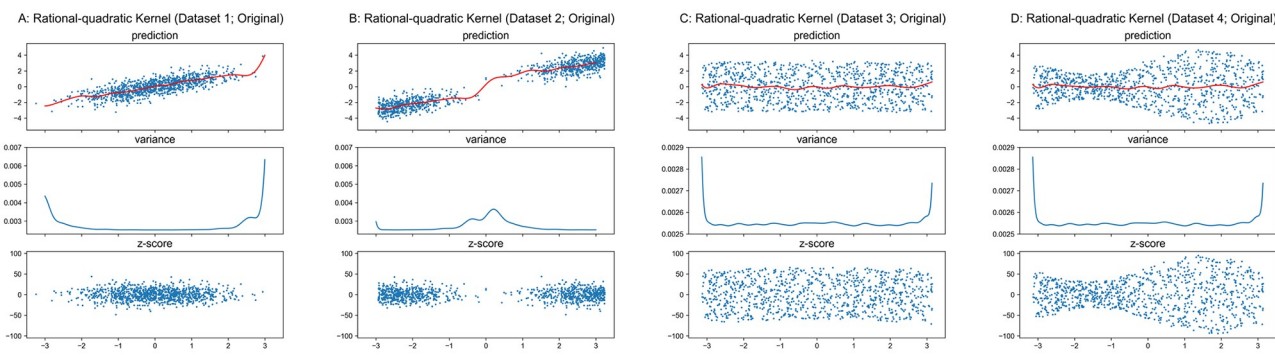

**Fig 10. GPR with Rational-Quadratic kernel on original datasets.** Scales of Y-axis for variance plots are different.

As the basic properties of linear and RBF kernels have been introduced, a hybrid kernel is utilized in the following analysis which is defined as

$$k_{\text{hybrid}}(\cdot, \cdot) = w_{\text{linear}}k_{\text{linear}}(\cdot, \cdot) + w_{\text{RBF}}k_{\text{RBF}}(\cdot, \cdot) + k_{\text{white}}(\cdot, \cdot), \tag{15}$$

where $w_{\text{linear}}$ and $w_{\text{RBF}}$ represent adjustable weights on linear and RBF kernels, and $k_{\text{white}}(\cdot, \cdot)$ refers to a white-noise kernel that represents the independently and identically normally-distributed observation noise, i.e., $\boldsymbol{K}_{\text{white}}(\boldsymbol{X}, \boldsymbol{X}) = \sum_{\text{train}}^{2}$, $\boldsymbol{K}_{\text{white}}(\boldsymbol{X}_*, \boldsymbol{X}_*) = \sum_{\text{test}}^{2}$, $\boldsymbol{K}_{\text{white}}(\boldsymbol{X}, \boldsymbol{X}^*) = \boldsymbol{0}$, and $\boldsymbol{K}_{\text{white}}(\boldsymbol{X}^*, \boldsymbol{X}) = \boldsymbol{0}$. Because the Matérn and Rational-Quadratic kernels are both based on the RBF kernel, so we only include the RBF kernel in the hybrid kernel. The Eq 5b can be reformulated as

$$\Sigma_*^2 = \boldsymbol{K}_{\text{hybrid}}(\boldsymbol{X}_*, \boldsymbol{X}_*) - \boldsymbol{K}_{\text{hybrid}}(\boldsymbol{X}_*, \boldsymbol{X})\boldsymbol{K}_{\text{hybrid}}(\boldsymbol{X}, \boldsymbol{X})^{-1}\boldsymbol{K}_{\text{hybrid}}(\boldsymbol{X}, \boldsymbol{X}_*). \tag{16}$$

Original Datasets 3–4 in Fig 2C and 2D are prefect for testing whether a model captures the heterogeneity $Var(y|\boldsymbol{x})$, as the large number of instances and uniformly distributed data over the input space lead to negligible epistemic uncertainty in certain input range, and the true reference model $y = 0$ is very simple as well. Besides, the results of two more complex datasets with quadratic reference models are presented in the S1 Appendix. In this section, the hyperparameters are tuned by maximizing the likelihood $P(\boldsymbol{y}|\boldsymbol{X}, \boldsymbol{\theta})$, where $\boldsymbol{\theta}$ represents all hyperparameters in the model. The results are plotted in Fig 12, and the optimized hyper-parameters are listed in Table 1 as well as the overall variances of residual $Var(y - y_{\text{reference}})$.

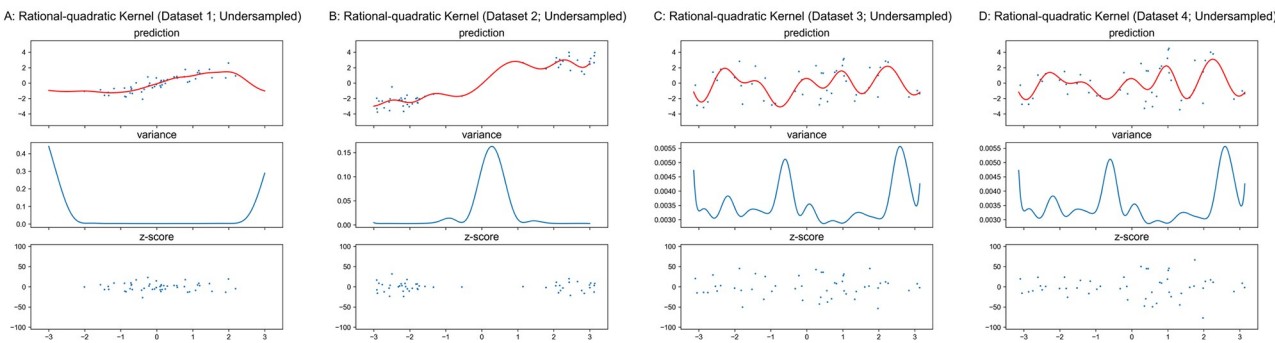

**Fig 11. GPR with Rational-Quadratic kernel on undersampled datasets.** Scales of Y-axis for variance plots are different.

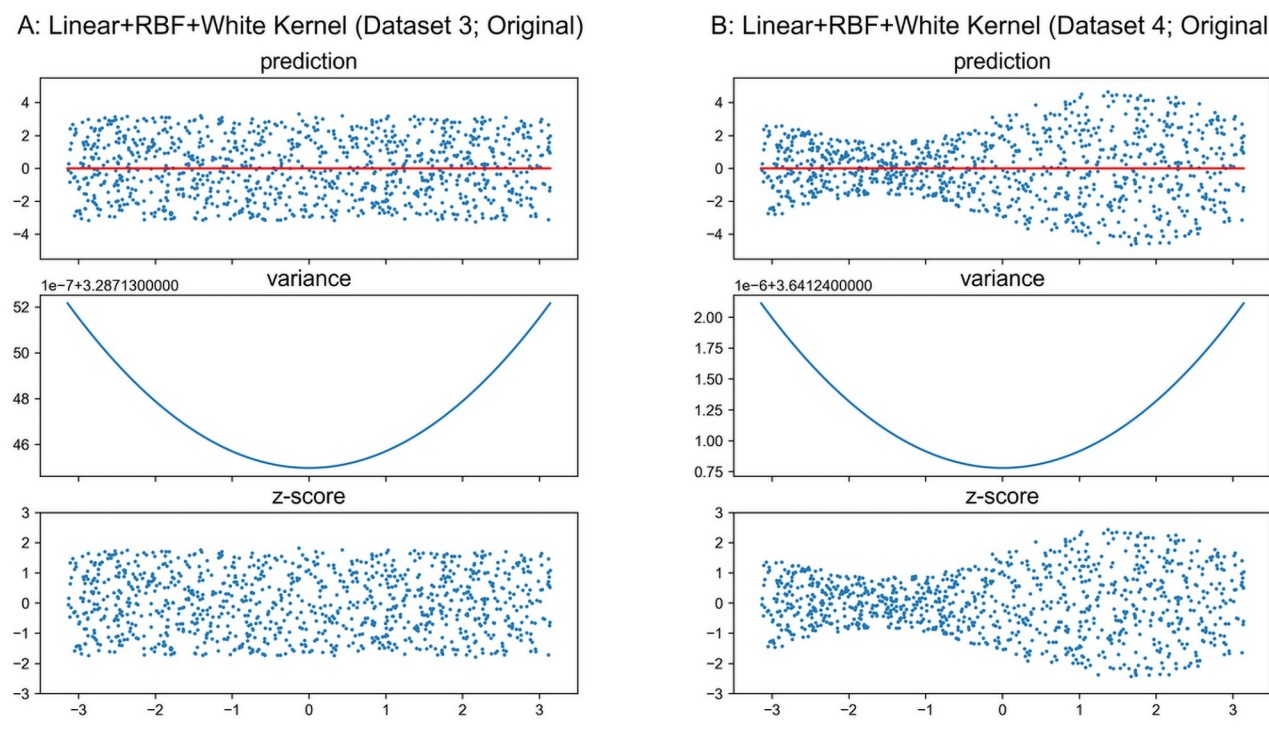

**Fig 12. GPR with hybrid kernel on original datasets 3–4.** Scales of Y-axis for variance plots are different.

As shown in Fig 12, the GPR accurately estimates the reference models, i.e., $y_{\text{reference}} \approx y_{\text{reference,true}}$. The variance curves are nearly quadratic, since the $w_{\text{linear}}$ is relatively larger than $w_{\text{RBF}}$ while $w_{\text{RBF}}$ is not exact zero as listed in Table 1. However, the domination of $k_{\text{white}}(\cdot, \cdot)$ over $k_{\text{linear}}(\cdot, \cdot)$ and $k_{\text{RBF}}(\cdot, \cdot)$ due to small optimized weights flattens the curves, i.e., the value of the curve is almost constant over the plotted input range in this example. Particularly, the $\sigma^2_{\text{noise}}$ is very close to the overall residual variance $Var(y-y_{\text{reference}})$, and the explanation will be presented later. Therefore, $\mathbf{K}_{\text{hybrid}}(\mathbf{X}_*, \mathbf{X}_*) \approx \sum^2_{\text{test}}$, $\mathbf{K}_{\text{hybrid}}(\mathbf{X}, \mathbf{X}) \approx \sum^2_{\text{train}}$, $\mathbf{K}_{\text{hybrid}}(\mathbf{X}^*, \mathbf{X}) \approx \mathbf{0}$ and $\mathbf{K}_{\text{hybrid}}(\mathbf{X}, \mathbf{X}^*) \approx \mathbf{0}$, which result in $\sum^2_* \approx \sum^2_{\text{test}} = \sum^2_{\text{noise}}$ for Eq 16.

Regarding Eq 1, since the noise is included as a tunable hyper-parameter without any constraints, the optimizer will adjust reference model $f(\cdot)$ as well as bias $\sigma^2_{\text{noise}}$ to $Var(y-y_{\text{reference}})$ to maximize the likelihood $P(\mathbf{y}|\mathbf{X},\boldsymbol{\theta})$. Even the $\sigma_{\text{noise}}$ refers to the observation noise level in GPR while the optimizer handles it as a variable without considering its meaning in a model.

In Dataset 3, $\sigma^2_{\text{noise}}$ is biased to the overall residual variance $Var(y-y_{\text{reference}})$, and $Var(y-y_{\text{reference}})$ is well matched with the homoskedastic heterogeneity $Var(y|\mathbf{x})$. So the $z$-scores plotted in Fig 12A show the GPR works as a normative model approach in this special case. However, in Dataset 4, $\sigma^2_{\text{noise}}$ is also biased to the overall residual variance $Var(y-y_{\text{reference}})$, while $Var(y-y_{\text{reference}})$ does not approximate the heteroskedastic

**Table 1. Optimized hyper-parameters for hybrid kernel on original datasets 3–4.**

| | $w_{\text{linear}}$ | $w_{\text{RBF}}$ | $l$ | $\sigma^2_{\text{noise}}$ | $Var(y-y_{\text{reference}})$ |
|---|---|---|---|---|---|
| **Dataset 3** | 7.29e-8 | 4.88e-10 | 2.94e2 | 3.29 | 3.29 |
| **Dataset 4** | 1.35e-7 | 1.98e-17 | 1.54e-5 | 3.64 | 3.64 |

heterogeneity $Var(y|\boldsymbol{x})$. So the $z$-scores plotted in Fig 12B do not represent a measure of normalized deviation in general.

## Discussion

Although GPR could be extended and to model the heterogeneity as presented in this work, it is either: (1) hard to estimate the aleatoric uncertainty accurately when the data are sparse, e.g., at the middle of Dataset 2; or (2) unnecessary to model the conditional variance by Eq 12 when the data are dense, e.g., Datasets 3–4. One approach to estimate $\sigma_{\text{aleatoric}}^2(\boldsymbol{x}_*)$ is using the sliding window technique, but it is hard to choose the window size for each dimension of input. For Scenario 1, even if the optimal window sizes can be obtained, it is hard to accurately estimate $\sigma_{\text{aleatoric}}^2(\boldsymbol{x}_*)$ when the window centered at $\boldsymbol{x}^*$ only covers a small number of training data points, e.g., $\boldsymbol{x}^*$ is far away from all points in $\boldsymbol{X}$. If the window centered at $\boldsymbol{x}^*$ covers a large number of training data points, e.g., Scenario 2, $Var(y|\boldsymbol{x}^*)$ should almost equal $\sigma_{\text{aleatoric}}^2(\boldsymbol{x}_*)$ and epistemic uncertainty is insignificant. Then $Var(y|\boldsymbol{x}^*)$ can be simply approximated as a local variance over a space defined by the window. There are more sophisticated algorithms than the naive sliding window technique, e.g., LOcal regrESSion (LOESS) [19, 20] and Generalized Additive Models of Location Shape and Scale (GAMLSS) [21, 22]. However, these methods still need densely distributed data over the input space based on our experience.

Another misunderstanding we found in the literature is interpreting the noise term $\sigma_{\text{noise}}^2$ as aleatoric uncertainty. When the observation noise is considered as a hyper-parameter, it will likely bias the overall residual variance $Var(y-y_{\text{reference}})$. The overall residual variance is a good approximation of homoskedastic aleatoric uncertainty $Var(y|\boldsymbol{x})$. It is, however, not valid for cases with heteroskedastic residuals, which is the main motivation for using normative modeling. Although the value of the noise term is biased to estimate overall residual variance during the optimization, the mathematical/physical meanings are pre-defined by the model. Moreover, in homoskedastic aleatoric uncertainty cases, further investigation is needed to verify whether $\boldsymbol{K}(\boldsymbol{X}_*, \boldsymbol{X}_*) - \boldsymbol{K}(\boldsymbol{X}_*, \boldsymbol{X})\left[\boldsymbol{K}(\boldsymbol{X}, \boldsymbol{X}) + \sum_{\text{noise}}^2\right]^{-1}\boldsymbol{K}(\boldsymbol{X}, \boldsymbol{X}_*)$ will still be a good approximation of epistemic uncertainty with such a biased estimation of observation noise level.

## Conclusions

In this paper, we present the mathematical derivation with a general formula to demonstrate that the derived prediction variance from GPR does not model the heterogeneity $Var(y|\boldsymbol{x})$, which in general is necessary for a normative model. GPR with a linear kernel and an RBF kernel are used as examples to illustrate this statement on one dimensional input datasets. Overall, the derived variance from GPR cannot be utilized in a normative model alone.

## Supporting information

**S1 Appendix. This file contains data/code availability, Eq S1, S1 Fig, and S1 Table.** All datasets are generated from the same Python script, which contains the code for analysis as well; Eq S1 is a detailed derivation for Eq 11; S1 Fig and S1 Table are results for modified Datasets 3-4.
(PDF)

## Acknowledgments

I would sincerely appreciate my families and friends from Laureate Institute for Brain Research, the University of Tulsa, and Brain Technologies Inc. for helping me through my

brain tumor surgery and the subsequent recovery during the revision of this paper. Special appreciation to Kaiping Burrows and Leandra Figueroa-Hall.

We would also like to thank the journal editor and anonymous reviewers for insightful discussions and feedback that have improved our study and manuscript.

## Author Contributions

**Formal analysis:** Bohan Xu.

**Investigation:** Bohan Xu.

**Methodology:** Bohan Xu.

**Writing – original draft:** Bohan Xu.

**Writing – review & editing:** Rayus Kuplicki, Sandip Sen, Martin P. Paulus.

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
